# Role of the Sortase A in the Release of Cell-Wall Proteinase PrtS in the Growth Medium of *Streptococcus thermophilus* 4F44

**DOI:** 10.3390/microorganisms9112380

**Published:** 2021-11-18

**Authors:** Ahoefa Ablavi Awussi, Emeline Roux, Catherine Humeau, Zeeshan Hafeez, Bernard Maigret, Oun Ki Chang, Xavier Lecomte, Gérard Humbert, Laurent Miclo, Magali Genay, Clarisse Perrin, Annie Dary-Mourot

**Affiliations:** 1CALBINOTOX, Université de Lorraine, F-54000 Nancy, France; ahoefa.awussi@gmail.com (A.A.A.); emeline.roux@univ-rennes1.fr (E.R.); zeeshan.hafeez@univ-lorraine.fr (Z.H.); arreteokc@gmail.com (O.K.C.); xavier.lecomte@qualtech-groupe.com (X.L.); gerard.humbert@univ-lorraine.fr (G.H.); Laurent.Miclo@univ-lorraine.fr (L.M.); magali.genay@univ-lorraine.fr (M.G.); clarisse.perrin@univ-lorraine.fr (C.P.); 2CNRS, LRGP, Université de Lorraine, F-54000 Nancy, France; catherine.humeau@univ-lorraine.fr; 3CNRS, Inria, LORIA, Université de Lorraine, F-54000 Nancy, France; bernard.maigret@univ-lorraine.fr; 4Hazard Substance Analysis Division, Gwangju Regional Office of Food and Drug Safety, Gwangju 10031, Korea

**Keywords:** *Streptococcus thermophilus*, sortase A, cell envelope proteinase, LPNTG motif, cell wall-anchoring

## Abstract

Growth of the lactic acid bacterium *Streptococcus thermophilus* in milk depends on its capacity to hydrolyze proteins of this medium through its surface proteolytic activity. Thus, strains exhibiting the cell envelope proteinase (CEP) PrtS are able to grow in milk at high cellular density. Due to its LPNTG motif, which is possibly the substrate of the sortase A (SrtA), PrtS is anchored to the cell wall in most *S. thermophilus* strains. Conversely, a soluble extracellular PrtS activity has been reported in the strain 4F44. It corresponds, in fact, to a certain proportion of PrtS that is not anchored to the cell wall but rather is released in the growth medium. The main difference between PrtS of strain 4F44 (PrtS_4F44_) and other PrtS concerns the absence of a 32-residue imperfect duplication in the prodomain of the CEP, postulated as being required for the maturation and correct subsequent anchoring of PrtS. In fact, both mature (without the prodomain at the N-terminal extremity) and immature (with the prodomain) forms are found in the soluble PrtS_4F44_ form along with an intact LPNTG at their C-terminal extremity. Investigations we present in this work show that (i) the imperfect duplication is not implied in PrtS maturation; (ii) the maturase PrtM is irrelevant in PrtS maturation which is probably automaturated; and (iii) SrtA allows for the PrtS anchoring in *S. thermophilus* but the SrtA of strain 4F44 (SrtA_4F44_) displays an altered activity.

## 1. Introduction

The lactic acid bacterium *S. thermophilus* is widely used in the manufacturing of domestic and industrial fermented dairy products [1], and has obtained the “Qualified Presumption of Safety” and “Generally Recognized as Safe” (GRAS) designations. It belongs to the *Streptococcus* genus which is mainly pathogenic or commensal [2,3].

*S. thermophilus* is auxotrophic for certain amino acids and needs to get them from its environment [4,5], whereas milk, the only known habitat of this bacterium, mainly contains proteins and very few immediately assimilable peptides and amino acids. Hence, *S. thermophilus* needs them for its growth to reach a high cell density of a functional surface proteolytic system [6,7]. It consists of the CEP PrtS, able to break down caseins into peptides, which are then internalized through specific transporters and hydrolyzed by cytoplasmic peptidases [8,9]. The *prtS* gene has been probably acquired from a bacterium close to *Streptococcus suis*, displaying the CEP SspA, which is very similar to PrtS (95% identities) [10,11].

PrtS is a subtilisin-like serine protease [12], synthetized and secreted as a pre-proproteinase. Its maturation remains unknown but could be achieved by peptidyl-prolyl isomerases [13]; as for the CEP PrtP and SpeB of *Lactococcus lactis* and *S. pyogenes*, they are maturated by PrtM and PrsA (homologous of PrtM), respectively [14,15]. PrtS is also usually anchored to the cell wall through its LPNTG motif located at its C-terminal extremity. This anchoring is thought to be achieved by the sortase A (SrtA), as the other surface proteins possess the LPXTG motif in Gram-positive bacteria. Sortases are ubiquitous in Gram-positive bacteria. Six classes (A to F) are distinguished, the most well-known being class A with SrtA of *Staphylococcus aureus* (for recent review, see [16]. In pathogen bacteria, sortases are then responsible for the anchoring of surface proteins involved in biofilm formation and/or their pathogenicity, and strategies of the inhibition of their activity are currently being developed [17,18].

After their synthesis as pre-proteins, the Sec-secretion machinery supports the LPXTG surface proteins of Gram-positive bacteria to export them through the cytoplasmic membrane. A signal peptidase cleaves the N-terminal signal peptide and proteins interact with the membrane by their C-terminal extremity. Then, the transpeptidase SrtA, thought to be located at the pole/septum with the Sec secretion system, anchors these proteins to the wall through LPXTG recognition [19,20]. Finally, it must be noted that (i) SrtA contains a non-cleaved signal peptide at its N-terminal extremity for both its exporting and holding onto the plasmic membrane, and (ii) the catalytic domain (triad Cys208, His142, and Arg216) is located in its C-terminal extremity.

Peptidyl-prolyl isomerases (PPIases, foldases, or maturases) maturate certain LPXTG-proteins, such as CEP [13], by catalyzing the cis-trans isomerization of a peptide bond located upstream a prolyl residue of the prodomain. In the case of the CEP SpeB of *S. pyogenes*, such a reaction has been postulated as essential for the stabilization of the pre-protein in a sec-pathway-dependent secretion-competent conformation; for its correct exportation; and for subsequent anchoring [21,22].

Chang et al. [23] detected two forms of PrtS in the strain 4F44: one anchored to the wall and an extracellular soluble one. Both forms are devoid of the 32 amino acids’ imperfect duplication in the prodomain PP, which is thought to be required for PrtS maturation [11], and the soluble form displays an intact LPNTG motif at its C-terminal extremity [23]. Two hypotheses were then proposed. The first implies that because of the absence of the 32 amino acids imperfect duplication in the PP domain, maturation of PrtS is incorrect, leading to an incorrect folding of the CEP and then to its imperfect anchoring, similarly to the SpeB of *S. pyogenes*. The second hypothesis relies on a partial deficiency of the anchoring activity of SrtA of strain 4F44 (SrtA_4F44_) in spite of only six substitutions between SrtA_4F44_ and SrtA_LMD-9_ (sortase A of the strain LMD-9). Indeed, one of the six, the Ile_218_ substitution, could be important for the recognition of the LPNTG motif [24]. Therefore, this study aims to address the cause of PrtS_4F44_ release in the extracellular medium.

## 2. Materials and Methods

### 2.1. Bacterial Strains and Culture Conditions

The strains and plasmids used in this study are presented in the Table 1. Strains were stored in reconstituted skim milk 10% (*m*/*v*) at −80 °C. They were precultured in reconstituted skim milk and then introduced at 1% into M17 medium with lactose 20 g L^−1^ (LM17) [25]. For the transformation experiments, the strains were first cultivated in LM17 before being inoculated in a chemically defined medium with 20 g L^−1^ of lactose [26]. The incubation temperature was 42 °C. For the LMD-9_∆*srtA*_, LMD-9_∆*prtS*_, and LMD-9_∆*prtM*_ mutants, 5 μg mL^−1^ of erythromycin was added to the LM17 medium, while for the mutants LMD-9*_srtA_*_LMD-9_, LMD-9*_srtA_*_4F44_, LMD-9*_prtS_*_4F44_, and LMD-9*_srtA_*_:Ile218__→Val218_, the medium was supplemented with 20 μg mL^−1^ of streptomycin and 300 μg mL^−1^ of spectinomycin. The growth of the strains was assessed for measuring the pH of the medium and the optical density (OD) at 600 nm.

### 2.2. DNA Extraction, PCR Amplification, Electrophoresis, and Sequencing Conditions

Plasmid DNAs were isolated from *Escherichia coli* using the Miniprep Kit (Fermentas, Villebon-sur-Yvette, France) according to the manufacturer’s instructions. Genomic DNAs were extracted as previously described [27]. Primer3plus software was used to design primers, which were then synthesized by Eurogentec (Seraing, Belgium). Sequences of primers and the sizes of amplicons are reported in Appendix A. Polymerase chain reactions (PCRs) were achieved according to the supplier’s recommendations (Fermentas, Villebon-sur-Yvette, France) in a Mastercycler proS thermocycler (Eppendorf, Hambourg, Germany). Cycle conditions were: 95 °C for 5 min, 35 cycles of 3 steps (95 °C for 30 s; hybridization at appropriate temperatures (Appendix A); 72 °C for 1 min kb^−1^), and 10 min at 72 °C. For mutant construction, high fidelity Phusion DNA polymerase (Fermentas, Saint Rémy-lès-Chevreuse, France) was used for the amplification of each fragment (final DNA concentration of 5 µg mL^−1^; extension time 30 s kb^−1^). For overlapping (OL) PCR, DNA fragments required for the constructions were pooled in equal amount (final concentration of 5 µg mL^−1^). The mixture also contained 0.5 µmol mL^−1^ of each primer (forward complementary with 5′ end of the first fragment and reverse complementary with 3′ end of the last fragment); high fidelity Phusion DNA polymerase 4U, each dNTP of 0.2 µmol mL^−1^; and 5X Phusion HF buffer of 4 µL. The program used was: 2 min at 95 °C, 35 cycles of 3 steps (95 °C 30 s, hybridization for 30 s at the annealing temperatures, and 72 °C 30 s kb^−1^), and finally 72 °C for 10 min.

The high pure PCR product purification kit (Roche Applied Science, Meylan, France) was used to purify PCR products, taking as an eluent either the elution buffer of the kit (for OL PCRs) or ultra-pure water (for sequencing reactions).

PCR products were separated by electrophoresis on 1% (*w*/*v*) agarose gel in 0.5× TAE buffer [28] at 100 V. The molecular weight markers used were 1 kb and 100 bp DNA ladders (Fermentas). Sequencing was performed by Beckman Coulter Genomics (Essex, UK) with the Sanger method [29].

### 2.3. Mutant Constructions and Natural Transformation

The *srtA* (locus: STER_RS06195; NCBI reference sequence: NC_008532.1) and *prtM* (locus: STER_RS02415; NCBI reference sequence NC_008532.1) genes of the *S. thermophilus* wild-type (WT) strain LMD-9 have been replaced by a cassette carrying the erythromycin resistance gene *ery*, leading to LMD-9_∆*srtA*_ and LMD-9_∆*prtM*_ mutants using the same strategy. Thus, the upstream (*prtM-*UpX or *srtA-*UpX) and downstream (*prtM-*DownX or *srtA-*DownX) fragments of gene *prtM* or *srtA* were amplified by PCR, as well as by the *ery* gene, which was located on plasmid pG+host 9. The three overlapped PCR amplicons (*prtM-*UpX–*ery–prtM-*DownX, or *srtA-*UpX*–ery–srtA-*DownX) obtained were used to create the recombinant fragment by overlapping PCR, which was then introduced into natural competent cells of *S. thermophilus* LMD-9, allowing for obtainment after its integration through a double crossing-over event, specifically either the LMD-9_∆*prtM*_ mutant or LMD-9_∆*srtA*_ mutant. Primers sequences used to amplify the different fragments are listed in Appendix A. The *ery* gene of mutant LMD-9_∆*srtA*_ had been replaced by the allele *srtA*_4F44_ (taxon: 1308; GenBank: GU459010.1) or *srtA*_LMD-9_, whereas the *ery* gene of mutant LMD-9_∆*prtS*_ had been replaced by the allele *prtS*_4F44_, leading to mutants LMD-9*_srtA4F44_*, LMD-9*_srtALMD-9_*, and LMD-9*_prtS4F44_*. For that, the upstream (UpX-*srtA* or UpX-*prtS*) or downstream (DownX-*srtA* or DownX-*prtS*) fragments of gene *srtA* or *prtS*, as well as the *srtA*_LMD-9_ gene, were amplified from the genomic DNA of the LMD-9 strain. The genes *srtA_4F44_* and *prtS_4F44_* were amplified from the genomic DNA of the 4F44 strain and the *spec* gene (conferring resistance to spectinomycin) was amplified from plasmid pSET4S. According to the mutant anticipated, four appropriate overlapped fragments (UpX-*srtA*, *srtA4F44*/*srtA_LMD-9_, spec*, DownX-*srtA* or UpX-*prtS**,* prtS*_4F44_*, *spec*, DownX-*prtS*) were used to create suitable recombinant fragments, which were then introduced into competent cells of mutant LMD-9_∆*srtA*_ or LMD-9_∆*prtS*_ to produce, after a double crossing-over event, the mutants LMD-9*_srtA4F44_*, LMD-9*_srtALMD-9_*, and LMD-9*_prtS4F44_*. Primer pairs used for the different amplifications are also indicated in Appendix A.

The strategy of the construction of mutant LMD-9*_srtA:Ile218_**_→Val218_* is presented in Figure A1 (Appendix B). Competent cells of LMD-9, LMD-9_∆*srtA*_, and LMD-9_∆*prtS*_ were prepared as described previously [26,31] to introduce, by natural transformation, the recombinant DNA fragments obtained by OL PCRs.

### 2.4. Detection of Extracellular Proteinase Activity

WT strains LMD-9 and 4F44, and mutants LMD-9_∆*srtA*_, LMD-9*_srtA_*_LMD-9_, LMD-9*_srtA_*_4F44_, and LMD-9*_srtA:Ile218_**_→Val218_* were cultured in 200 mL of LM17 until OD_600 nm_ = 1. Supernatants of centrifuged (3900 *g*–10 min) cultures were filtered and then stored at 4 °C. Bacterial pellets were washed in Tris-HCl buffer (100 mmol L^−1^, pH 7), then resuspended in 20 mL of the same buffer and kept at 4 °C. The free (supernatant) or bound (pellet) PrtS activity was revealed using the Suc-Ala-Ala-Pro-Phe-*p*NA substrate (Sigma, St. Louis, MO, USA) [23]. PrtS activity was expressed in arbitrary units, each measure being realized at least in triplicates. Activity values greater than one can be compared since they were established during the same assay and therefore in the same experimental condition for the strains LMD-9*_prtS4F44_* and LMD-9_∆*prtS*_; LMD-9 and LMD-9_∆*prtM*_; LMD-9_∆*srtA*_, LMD-9, and LMD-9*_srtALMD-9_*; and LMD-9*_srtA4F44_* and 4F44. (Table 1) The free activity of PrtS was also demonstrated by zymography with caseins as substrates [7].

Molecular modeling simulations were run on a bi-processor AMD Dual Core 280 with 2.4 GHz. Docking and scoring simulations were performed using the LibDock algorithm [34] and the Consensus Score modules of the program-package Discovery Studio version 3.5 (Accelrys, Inc., San Diego, CA, USA), respectively. All molecular mechanics calculations were performed with the CHARMm force field [35]. The Protein Data Bank entry 3FN5 corresponding to the SrtA (Spy1154) of the *S. pyogenes* serotype M1 strain SF370 was used as the input structure [24]. This was made of two chains per unit cell (length: a—39.8 Å, b—59.46 Å, and c—65.11 Å; angles: α—90°, β—101.96°, and γ—90°). These two chains contained 187 residues corresponding to the catalytic domain. The first 18 residues corresponded to the His-tag region expediting the purification of the protein. The chain A included a 4-(2-Hydroxyethyl)-1-Piperazine Ethanesulfonic Acid (HEPES) entity as an inhibitor. The chain B was picked as the input structure for all simulations. In our study, the first 18 residues due to the crystallization step were withdrawn from chain B. The resulting structure Sp-SrtA_SF370∆86_ was used as a template to design homology models for St-SrtA_4F44∆90_ and St-SrtA_LMD-9∆90_. In order to investigate the proteinase PrtS “sorting pattern” binding modes within the SrtA active site, two ligands were designed, namely LPNTG and Ace-QLPNTGEND-NMe. Their possible binding modes within the active site of the SrtA systems (Sp-SrtA_SF370∆86_, St-SrtA_4F44∆90_, and St-SrtA_LMD-9∆90_) were studied through docking simulations using the LibDock algorithm. This method allows the identification of low-energy binding modes of ligands based on polar and apolar interactions sites (hotspots). The SrtA hotspots were defined as the catalytic site which encloses the catalytic residues. To prevent potential interactions, water molecules were removed from the cavity. The structure of the ligand complies following polar and apolar interaction sites of the receptor. Using this docking procedure guarantees that only the highest scoring poses (30 to 100 poses) are kept. The Libdock scoring functions based on simple pair-wise score calculations were used in all simulations. Finally, poses were assessed considering the position and orientation of the sorting pattern within the catalytic cavity and its proximity to the catalytic triad.

## 3. Results and Discussion

### 3.1. The Imperfect Duplication of 32 Amino Acid Residues in the Prodomain of PrtS_LMD-9_ Is Not Involved in the Anchoring of PrtS to the Cell Wall of S. thermophilus

Delorme et al. [11] postulated that the imperfect duplication of 32 amino acid residues (residues 63 to 90) in the prodomain PP of PrtS of the strain LMD-9 (PrtS_LMD-9_; locus: STER_RS04165; NCBI reference sequence: NC_008532.1) is responsible for the anchoring and/or maturation of this CEP. To evaluate this, the LMD-9*_prtS_*_4F44_ mutant strain was constructed by integrating the *prtS* allele of the 4F44 strain (*prtS*_4F44_ (taxon: 1308; GenBank: GU459009.1)), encoding a CEP devoid of such duplication, in the LMD-9_∆*prtS*_ mutant strain [31]. The proteolytic activity was evaluated at the cells’ surface and in the growth supernatant of this mutant as well as the mutant LMD-9_∆*prtS*_, served as a negative control. Followed by the growth of both strains in LM17 to an OD_600nm_ of 1, cells were harvested by centrifugation and both the filtered supernatants and the cells were subsequently incubated with the synthetic substrate Suc-Ala-Ala-Pro-Phe-*p*NA. As anticipated, a strong proteolytic activity was noticed at the surface of the cells of LMD-9*_prtS_*_4F44_ but not in the filtered supernatant, whereas no such activity was reported for the negative control (Table 1). Hence, it is concluded that any mutation in the sequence of PrtS_4F44_, particularly the absence of the imperfect duplication of 32 amino acid residues in its prodomain PP, is not responsible for the liberation of PrtS_4F44_ into the extracellular medium of strain 4F44.

### 3.2. Role of Maturases in the Maturation of PrtS in S. thermophilus LMD-9

An exploration of data banks allowed for identification of four genes that encode PPIases (*prtM*, *tig* or *ropA*, *ppiA*, and *pplB*) in sequenced genomes of *S. thermophilus.* Among them, two maturases, PplB and PpiA, belong to the cyclophilin family, whereas PrtM and RopA belong to the parvulin and FKBP (FK506 binding protein) families, respectively. An analysis of these four genes in the 4F44 strain revealed that their deduced proteins are identical with the corresponding ones of the LMD-9 strain, except the PrtM protein (taxon: 1308; locus: AMM43147; GenBank: KT809299.1), which displays four differences (D30N, V75A, A78V, and A242T). Hence, to determine its implication in PrtS maturation, the LMD-9_∆*prtM*_ mutant strain was constructed by deleting the *prtM* gene in the LMD-9 strain.

The LMD-9, LMD-9_∆_*_prtS_*, and LMD-9_∆_*_prtM_* strains were grown in milk to evaluate the effect of *prtM* deletion on the growth performance of the mutant. This approach is an indirect measure of the PrtS proteolytic activity considering the relationship between the PrtS activity and *S. thermophilus* capacity to grow in milk [6,7]. The results revealed that the LMD-9_∆_*_prtS_* mutant showed a delayed growth both in LM17 and milk media contrary to strains LMD-9 and LMD-9_∆_*_prtM_*, which showed a similar growth behavior (Figure 1). In a similar manner, the proteolytic activity was observed only at the cells’ surface of LMD-9 and LMD-9_∆_*_prtM_* (Table 1), and not in the culture supernatants.

Consequently, the lack of the putative PrtM maturase in the LMD-9_∆_*_prtM_* strain did not result in any growth retardation, nor in a decrease in PrtS activity and/or its release into the extracellular medium; it is most probable that the PrtM maturase was not responsible for the PrtS maturation. Several factors support the hypothesis that PrtS could undergo automaturation. First, unlike thee *prsA* of *S. pyogenes* and *prtM* of *L. lactis*, which are located upstream of the *speB* and *prtP* genes, respectively and co-transcribed, the *prtM* of *S. thermophilus* is not located near PrtS [9,12,14,36]. Second, when Chang et al. [23] established the N-terminal sequence of the soluble PrtS form of the 4F44 strain, they detected the N-terminal sequence corresponding to the proenzyme (with prodomain PP) as well as the N-terminal sequence of the mature form (without prodomain PP) of PrtS. The same situation was also detected for anchored PrtS forms present at the cells’ surface of the LMD-9 strain [31]. It was also noticed that the PrtS proenzyme form disappeared progressively in favor of the mature form (unpublished results) preceding a prolonged incubation or high concentration preservation. The fact that PrtS could be automaturated is not inimitable in LAB, e.g., regarding CEP PrtB of *Lb. bulgaricus* [37]. Finally, the fact that the inactivation of *prtM* did not lead to any release of PrtS in the growth medium, associated with the absence of difference between the three other known maturases (PpiA, PplB, and RopA), ruled out the hypothesis of a link between maturation via PrtM especially, correct folding, and anchoring.

### 3.3. SrtA Is Responsible for the Anchoring of PrtS to the Cell Wall of S. thermophilus and Is Deficient in Strain 4F44

The hypothesis that, despite few numbers of substitutions found between SrtA_4F44_ and SrtA_LMD-9,_ the extracellular liberation of PrtS_4F44_ could result from a partial deficiency of SrtA_4F44_ implies first providing evidence that SrtA is actually responsible for the PrtS anchoring to the cell wall of *S. thermophilus* as in other Gram-positive bacteria. Hence, the LMD-9_∆*srtA*_ mutant strain was constructed by replacing the *srtA* gene by an erythromycin resistance gene. Afterwards, a complemented mutant was constructed by reintegrating the *srtA*_LMD-9_ gene to the LMD-9_∆*srtA*_ mutant to verify whether the phenotype of wild type LMD-9 strain could be restored (LMD-9*_srtA_*_LMD-9_).

The synthetic substrate Suc-Ala-Ala-Pro-Phe-*p*NA was used to search proteolytic activity in the filtered growth medium of the LMD-9_∆*srtA*_ mutant, complemented mutant, and WT strain (Table 1). It was detected only in the LMD-9_∆*srtA*_ mutant (1020 mAU of PrtS activity; Table 1). The same supernatants were then analyzed by a casein-zymogram to detect PrtS (Figure 2A). Three caseinolytic bands (170, 154, and 115 kDa) previously shown to correspond to PrtS [23] were observed (Figure 2A) in the supernatant of mutant LMD-9_∆*srtA*_ contrary to WT and LMD-9*_srtA_*_LMD-9._ This corroborates the results obtained using synthetic substrate Suc-Ala-Ala-Pro-Phe-*p*NA. Unexpectedly, the LMD-9_∆*srtA*_ strain displayed a surface PrtS activity (Table 1) despite of *srtA* deletion. To determine whether this resulted from electrostatic or other low-force interactions of PrtS with the cell surface, cells of this mutant and of strains 4F44 and LMD-9*_srtA_*_LMD-9_, used as controls, were suspended in Tris HCl (100 mmol/L, pH 7) buffer and incubated from 0 to 70 h at 4 °C. The cells’ surface-bound PrtS activity was determined at t0 and t70 using a casein-zymogram (Figure 2) and the synthetic substrate Suc-Ala-Ala-Pro-Phe-*p*NA. After 70 h of incubation, the surface PrtS activity of strains 4F44, LMD-9, and LMD-9*_srtA_*_LMD-9_ was close to that obtained at t0, while that of the LMD-9_∆*srtA*_ strain appeared to be undetectable. The PrtS activity was exclusively found in the incubation buffer of this strain (Figure 2C). These last results proved that the SrtA of *S. thermophilus* is responsible for the anchoring of PrtS to the cell wall and probably the other proteins possessing a LPXTG motif. Such a phenotype has already been described in a *St. aureus* mutant that was defective in the anchoring of surface LPXTG proteins because of a mutation in the *srtA* gene. The deletion of this gene resulted in the liberation of surface LPXTG proteins, thereby leading to a decreased virulence of the bacterium [38]. In a similar manner, in the *Streptococcus* genus, another study demonstrated that SrtA is a key player responsible for the anchoring of surface LPXTG proteins of *S. pyogenes* [39].

To further investigate whether SrtA_4F44_ is partially defective, the LMD-9*_srtA_*_4F44_ mutant was constructed by introducing the *srtA_4F44_* allele into the genome of the LMD-9_∆*srtA*_ strain. Hereafter, the soluble PrtS activity was examined in filtered growth supernatants of the strains LMD-9*_srtA_*_4F44_ and 4F44 by using both the Suc-Ala-Ala-Pro-Phe-*p*NA substrate (Table 1) and a casein-zymogram (Figure 2B). Through analyses being realized in the conditions, levels of PrtS activity could be compared (see Section 2). Twenty-six percentages of total PrtS activity of the mutant LMD-9*_srtA_*_4F44_ were found in its growth supernatant, i.e., a proportion similar to that of strain 4F44 (37%). In addition, the incubation of the cells of this mutant in Tris-HCl (100 mmol L^−1^, pH 7) buffer during 70 h did not result in a significant increase of extracellular PrtS activity. Therefore, despite the few numbers of substitutions found between the SrtA of strains LMD-9 and 4F44, our results provide genetic proof of the dysfunctioning of SrtA_4F44_, which leads to the anchoring to the cell wall of the majority of PrtS molecules and to the release of a fraction of PrtS in the growth medium of the 4F44 strain, as postulated by Chang et al. [23]. Indeed, the fact that after 70 h of incubation of the cells of 4F44 and LMD-9*_srtA4F44_* strains in Tris-HCl buffer the surface PrtS activity remained similar to the initial one strongly suggests that it corresponds to PrtS molecules correctly anchored to the cell wall. Besides, even if the total PrtS activity (bound plus free) appeared in our assays to be higher in strain 4F44 than in strain LMD-9 (8860 mAU against 4194, Table 1), the PrtS release cannot be attributed to a higher expression of its gene in strain 4F44, leading to the saturation of SrtA activity and ultimately to a leakage of non-anchored PrtS molecules in the external environment. Indeed, in the mutant LMD-9*_srtA4F44_*, the gene *prtS* undergoes the same regulation like in the wild-type LMD-9 strain, as suggested by the PrtS activity levels observed in this mutant (Table 1). Therefore, no saturation of the anchoring activity of sortase SrtA is expected and SrtA_4F44_, which is expressed in this mutant, should anchor all PrtS molecules.

### 3.4. Substitution of the Ile 218 Residue Is Not Responsible for the Deficiency of SrtA_4F44_

As we knew the LPNTG motif was present at the C-terminal extremity of the extracellular soluble form of PrtS_4F44_ [23], molecular modeling simulations were performed to determine whether the binding mechanism of the LPNTG motif to the SrtA catalytic site was at least partially altered in SrtA_4F44_.

Since *S. pyogenes* and *S. thermophilus* belong to the same genus, their respective sortases A were assumed to have similar mechanisms. Thus, the structural models of SrtA_LMD-9_ (St-SrtA_LMD-9∆90_) and SrtA_4F44_ (St-SrtA_4F44∆90_) were built from the already defined structure of *S. pyogenes* SF370 SrtA (Sp-SrtA_SF370∆86_) [24]. In order to build the models with the same amino acid residues as in the Sp-SrtA_SF370∆86_, it was necessary to delete the first 90 residues of St-SrtA_LMD-9∆90_ and SrtA_4F44_ (St-SrtA_4F44∆90_). The percentage of identity/similarity between the C-terminal domain of Sp-SrtA_SF370∆86_ and St-SrtA_4F44∆90_, and Sp-SrtA_SF370∆86_ and St-SrtA_LMD-9∆90_ were found to be sufficient (between 70.6% to 97.5% and 87.1% to 99.4%, respectively) to use Sp-SrtA_SF370∆86_ as a structural pattern for constructing the St-SrtA_4F44∆90_ and St-SrtA_LMD-9∆90_ ones. Hence, surimposition of the three structures led to RMSD values below 0.16 Å considering all the 169 C_α_ atoms of the residues 87 to 249. No significant structural difference was observed between the structures, as St-SrtA_4F44∆90_ and St-SrtA_LMD-9∆90_ models displayed the characteristic structure of sortases, i.e., the eight-stranded β-barrel fold and a long hydrophobic cleft corresponding to the catalytic cavity located at the center of the protein (Figure 3). It was assumed that the residues Cys_208_, His_142_, and Arg_216_ compose the catalytic triad (Figure 3). The orientation of these residues is consistent with the model of reverse protonation that has been proposed in biochemical studies of the sortase A of *St. aureus* and *S. pyogenes* [24,40].

These homology models were then used to study possible binding modes of the LPNTG pattern within the sortase catalytic cavity. Enzyme/substrate complexes were generated using docking simulations with two substrates: the LPNTG pattern and a longer pattern, specifically the Ace-QLPNTGEND-Nme pattern. Unfortunately, no significant difference was observed between the complexes Sp-SrtA_SF370∆86_: LPNTG/Ace-QLPNTGEND-Nme; St-SrtA_LMD-9∆90_: LPNTG/Ace-QLPNTGEND-Nme; and St-SrtA_4F44∆90_: LPNTG/Ace-QLPNTGEND-Nme (Appendix A and Figure A2 and Figure A3 in Appendix B). Consequently, these results strongly suggested that none of the four substitutions distinguishing St-SrtA_4F44∆90_ from St-SrtA_LMD-9∆90_ could account for the defective activity of the SrtA_4F44_.

To definitively rule out the hypothesis of an eventual role of the substitution of Ile_218_ by Val_218_ in the deficiency of StrA_4F44_ and to validate the modeling prediction, the residue Ile_218_ of strain LMD-9*_srtA_*_LMD-9_ was replaced by a valine residue (Figure A1). The absence of extracellular PrtS activity (searched using the Suc-Ala-Ala-Pro-Phe-*p*NA substrate) in the supernatant of this strain confirms the non-involvement of Ile_218_ substitution by Val_218_ in the deficiency of StrA_4F44_.

To conclude we showed that (i) the 32 amino acid residues’ imperfect duplication located in the prodomain of certain PrtS, such as PrtS_LMD-9_, was not essential for the correct maturation and subsequent anchoring of PrtS; (ii) the maturase PrtM, homologous to maturases of other CEP, was not responsible for the maturation of PrtS and neither for its correct anchoring to the cell surface; and (iii) SrtA was responsible for the anchoring of PrtS to the cell wall and (iv) SrtA of strain 4F44 was partially defective despite the low number of dissimilar residues (six substitutions), which differentiates it from that of the LMD-9 strain and probably through a subtle mechanism not yet elucidated perhaps because of a lack of a structural model, including the N-terminal part of SrtA.

## Figures and Tables

**Figure 1 microorganisms-09-02380-f001:**
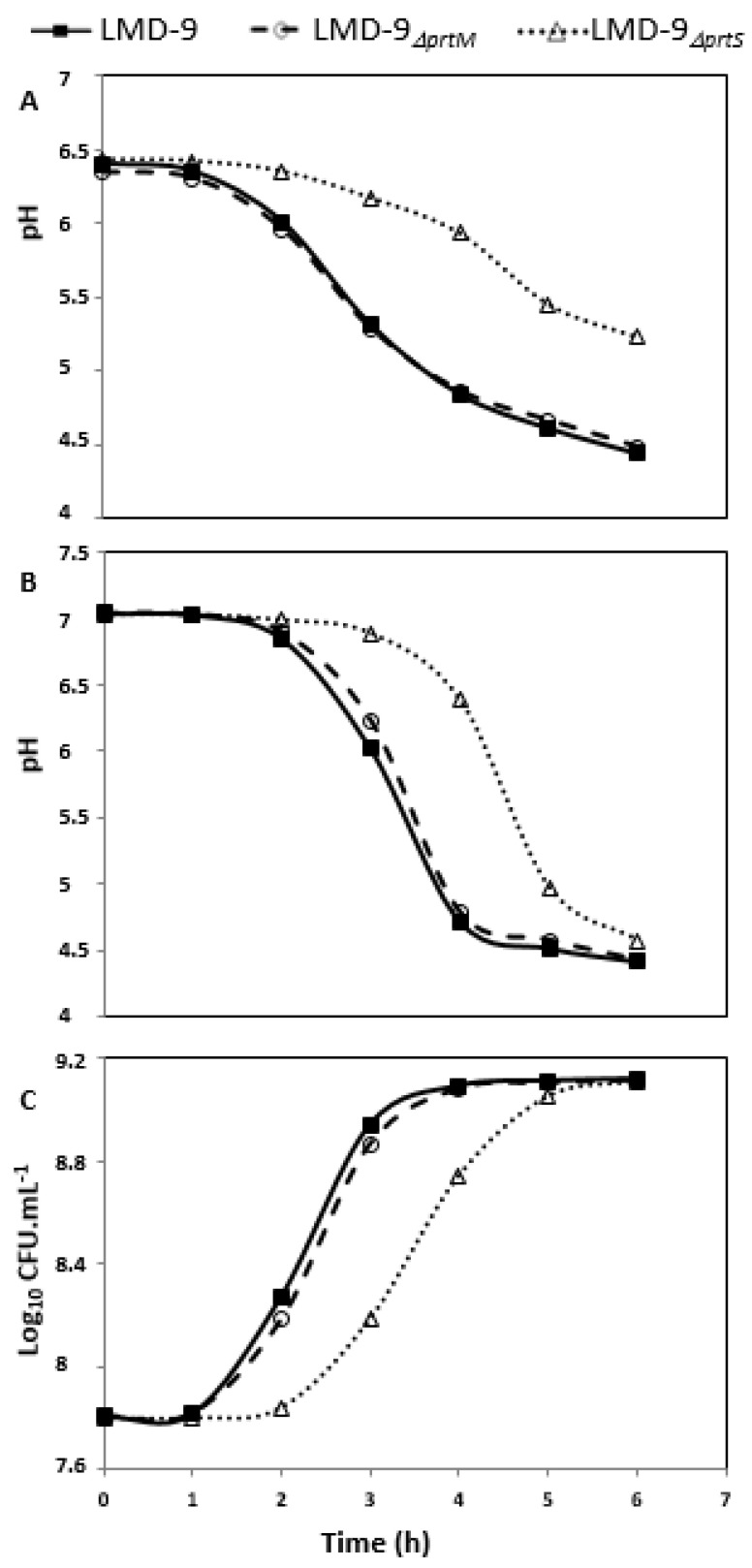
Growth of *S. thermophilus* LMD-9, LMD-9_∆*prtM*_, and LMD-9_∆*prtS*_ (a negative control) in milk (**A**) or in LM17 (**B**,**C**), evaluated by measuring either the extracellular pH (**A**,**B**) or OD_600_ (**C**).

**Figure 2 microorganisms-09-02380-f002:**
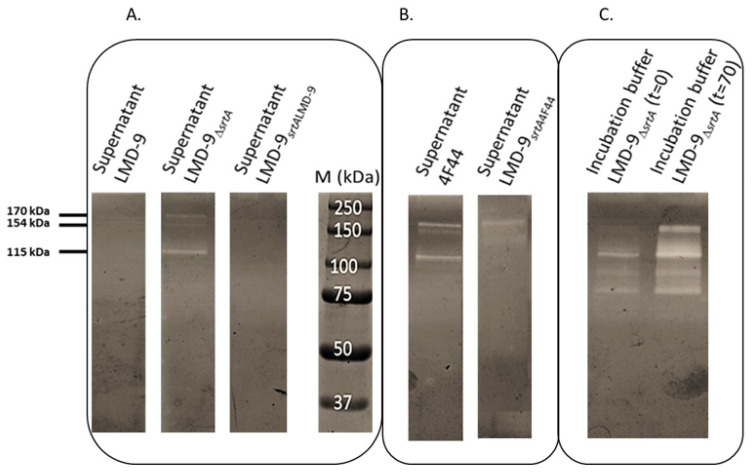
SDS-PAGE zymogram analysis to detect proteolytic activity in the extracellular growth medium of *S. thermophilus* LMD-9, LMD-9_∆_*_srtA_*, LMD-9*_srtA_*_LMD-9_ (**A**), and 4F44, LMD-9*_srtA_*_4F44_ (**B**), and the proteolytic activity liberated in the incubation buffer (t = 0 and t = 70 h) of the LMD-9_∆_*_srtA_* strain (**C**).

**Figure 3 microorganisms-09-02380-f003:**
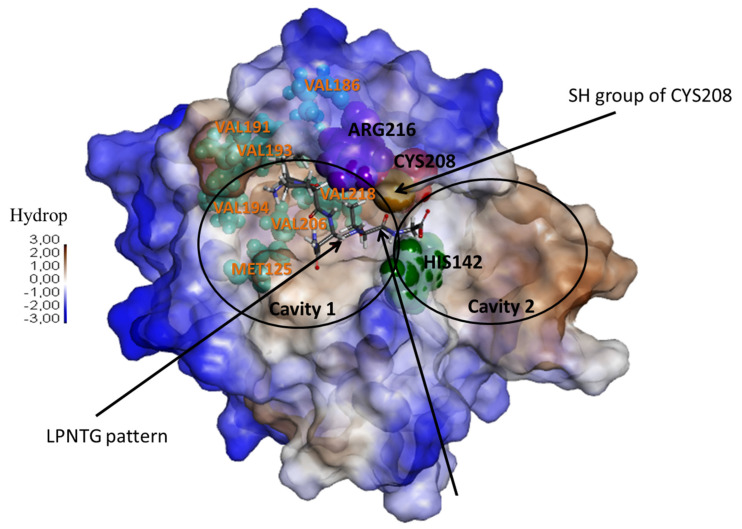
Putative structure of St-SrtA_4F44∆90_. The catalytic triad (in CPK, His_142_ in green, Arg_216_ in blue, and Cys_208_ in red, and its thiolate group in brown) position and the amino acid residues important (according to Race et al. [24]) for the positioning of the LPNTG motif within the cavity (in CPK teal) are indicated. The LPNTG motif is also shown within the cavity of SrtA_4F44∆90_. The structure of Sp-SrtA_SF370∆86_ and St-SrtA_LMD-9∆90_ are analogous to that of 4F44. The residue Val_218_ corresponds to the residue Ile_218_ in St- SrtA_LMD-9∆90_, while the residues Val_194_ and Val_218_ correspond to residues Ile_214_ and Ile_218_ in Sp-SrtA_SF370∆86_. For simplicity, the catalytic cavity has been divided into cavity 1 and cavity 2.

**Table 1 microorganisms-09-02380-t001:** Characteristics of *Streptococcus thermophilus* strains and plasmids used in this study and PrtS activity.

Strains\Plasmids	Origin	References	*srtA* Allele	Presence (+)/Absence (−) of the *prtS* Gene	PrtS Activity in mAU	
					Cell Surface	Growth Supernatant
*Streptococcus thermophilus* wild-type						
LMD-9	Yogurt	[30]	*srtA* _LMD-9_	+	3110	0
4F44	Cheese	[23]	*srtA* _4F44_	+	5620	3240
*Streptococcus thermophilus* mutants						
LMD-9*_srtA_*_LMD-9_	LMD-9	Present study	*srtA* _LMD-9_	+	3100	0
LMD-9_∆_*_srtA_*	LMD-9	Present study	Δ	+	2600	1020
LMD-9*_srtA_*_4F44_	LMD-9	Present study	*srtA* _4F44_	+	3124	1070
LMD-9_∆_*_prtS_*	LMD-9	[31]	*srtA* _LMD-9_	−	0	0
LMD-9*_prtS_*_4F44_	LMD-9	Present study	*srtA* _LMD-9_	+	3093	0
LMD-9_∆_*_prtM_* LMD-9*_srtA_*_:Ile218__→__Val218_	LMD-9 LMD-9	Present study Present study	*srtA*_LMD-9_ *srtA*:_Ile218__→__Val218_	+ +	3114 Present	0 0

Plasmids			Relevant markers and characteristics			
pG+host9	-	[32]	Erm^R^ *, pWV01 derivative, with thermoresistant replication function			
pSET4s	-	[33]	Spec^R^ *, replication function of Pg + host3 and pUC19			

* Spec^R^ and Erm^R^: resistance to spectinomycin and erythromycin, respectively; Δ: deletion of the *srtA*, *prtS*, or *prtM* gene; and mAU: milli arbitrary units.

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
