# Peer review of "Role of the Sortase A in the Release of Cell-Wall Proteinase PrtS in the Growth Medium of Streptococcus thermophilus 4F44"

_microorganisms, 2021, doi:10.3390/microorganisms9112380_

Round 1
Reviewer 1 Report
The manuscript “Role of the sortase A SrtA in release of cell-wall proteinase PrtS in growth medium of Streptococcus thermophilus 4F44” (ID 1427343) reports incidence of sortase SrtA on anchoring and release of cell envelope proteinase PrtS in different S. thermophilus strains. The authors have presented detection of protease activity in supernatant and cell surface by detection of hydrolysis of chromogenic substrate and zymography using casein in mutants ∆srtA, ∆prtM or ∆prtS and with gene replacements. Authors have also compared structural model of StrA to established structure of StrA of S. pyogenes in order to establish differences. Authors have shown that StrA is involved in anchoring of PrtS to cell wall while imperfect duplication in the PrtS prodomain and the maturase PrtM did not in maturation and anchoring. StrA has been shown partially deficient for the anchoring PrtS in strain 4F44 but the mechanism is yet unknown.
There is a range of comments that need to be addressed.
Minor comments
- line 269 : “was constructed” is in italic.
- line 291 : “a” is in italic.
Major comments :
- Figure 1 should be deleted to the introduction. This figure corresponding to protein alignment was already described as Figure 3 in the previous paper Chang et al., International Dairy Journal 2017.
- Figure A1 is not useful and should be deleted. Explanations described in the Figure legend should be introduced in the Materials and Methods (2.3 Mutant constructions and natural transformation).
- Figure A2 needs to be simplified or deleted since different steps in gene replacement are now well known. Explanations in the Figure legend are very clear and should be introduced in the Materials and Method (2.3 Mutant constructions and natural transformation).
- Authors should clarify the second point that support the hypothesis about the auto-maturation of PrtS. What did they mean exactly by presence of pro and mature form of PrtS in the 4F44 strain that supports this hypothesis (line 248-251)?
- Authors detected protease activity using chromogenic substrate. Protease activity is expressed as arbitrary unit corresponding to an absorbance determined by spectrometry. There is no calibration range to obtain reliable quantification. We have no idea about stability, reproducibility and detection limits of these assays. How many biological replicates have been done? In this context, protease activity is detected but it is not possible to compare activity between strains. Authors should replace line 305-307 and line 316-320 that presents percentage and proportion.
- Supplementary data should be introduced in results section and probably in Fig. A4 and Fig. A5 legends.
- Crystal structure and peptide substrate preference of SrtA from pneumonia have been recently published (Biswas T. et al., 2020 Biochem. J. 477 (24) : 4711-4728). Can this Sp StrA structure be used to compare to StrA from S. thermophilus and found other amino acids substitution or structural differences ?
Reviewer 2 Report
In this study the authors examined the role of the sortase A in release of cell-wall proteinase PrtS 2 in growth medium of Streptococcus thermophilus 4F44.
The study is well designed and very specific; it will add more to current evidence, however, few points need to be addressed.

Author Response
Please look at the attached file.
